# A portfolio approach to massively parallel Bayesian optimization

## Abstract

One way to reduce the time of conducting optimization studies is to evaluate designs in parallel rather than just one-at-a-time. For expensive-to-evaluate black-boxes, batch versions of Bayesian optimization have been proposed. They work by building a surrogate model of the black-box to simultaneously select multiple designs via an infill criterion. Still, despite the increased availability of computing resources that enable large-scale parallelism, the strategies that work for selecting a few tens of parallel designs for evaluations become limiting due to the complexity of selecting more designs. It is even more crucial when the black-box is noisy, necessitating more evaluations as well as repeating experiments. Here we propose a scalable strategy that can keep up with massive batching natively, focused on the exploration/exploitation trade-off and a portfolio allocation. We compare the approach with related methods on noisy functions, for mono and multi-objective optimization tasks. These experiments show orders of magnitude speed improvements over existing methods with similar or better performance.

## 1 Introduction

Current trends in improving the speed or accuracy of individual computations are on exploiting parallelization on highly concurrent computing systems. These computer models (a.k.a. simulators) are prevalent in many fields, ranging from physics to biology and engineering. Still, increasing the parallelization for individual simulators often comes with diminishing returns and model evaluation time remains limiting. A strategy is then to conduct several evaluations simultaneously, in batches, to optimize (here minimize) quantities of interest. See, e.g., (Haftka et al., 2016) for a review.

For fast simulators, evolutionary algorithms (EAs) are amenable to parallelization by design, see e.g., the review by Emmerich and Deutz (2018). But they require a prohibitive number of evaluations for more expensive-to-evaluate simulators. For these, Bayesian optimization (BO), see e.g., (Shahriari et al., 2016; Frazier, 2018; Garnett, 2022), is preferred, with its ability to carefully select the next evaluations. Typically, BO relies on a Gaussian process (GP) model of the simulator, or any black-box, by using a probabilistic surrogate model to efficiently perform the so-called exploration/exploitation trade-off. Exploitation refers to areas where the prediction is low (for minimization), while exploration is for areas of large predictive variance. An infill criterion, or acquisition function, balances this trade-off to select evaluations, such as the expected improvement (EI) (Mockus et al., 1978) in the efficient global optimization algorithm (Jones et al., 1998). Alternatives include upper confidence bound (UCB) (Srinivas et al., 2010), knowledge gradient (Frazier, 2018), and entropy based criteria (Villemonteix et al., 2009b; Hennig and Schuler, 2012; Wang and Jegelka, 2017). Parallelization is then enabled by the definition of batch versions of the corresponding infill criteria, selecting several designs to evaluate at once.

Noisy simulators have their own set of challenges, see e.g., (Baker et al., 2022), and raise questions about selecting the right amount of replication. While not necessary per se, repeating experiments is the best option to separate signal from noise, and is beneficial in terms of computational speed by limiting the number of unique designs, see e.g., (Binois et al., 2019; Zhang et al., 2020). Here, we also consider multi-objective optimization (MOO) where the goal is to find the set of best compromise solutions, the Pareto front, since

there is rarely a solution minimizing all objectives at once. We refer to, e.g., (Hunter et al., 2017) for a review of MOO options for black-boxes and (Emmerich et al., 2020) for multi-objective (MO) BO infill criteria. MO versions of batch algorithms have also been proposed, taking different scalarization weights (Zhang et al., 2010), or relying on an additional notion of diversity (Lukovic et al., 2020).

Our motivating example is the calibration of a large-scale agent-based model (ABM) of COVID-19 run on a supercomputer (Ozik et al., 2021) with the added goal of reducing the *time to solution* for meaningful, i.e., timely, decision-making support. The targeted setup is as follows: a massively parallel system (e.g., HPC cluster or supercomputer) with the ability to run hundreds of simulation evaluations in parallel over several iterations for the purpose of reducing the overall time to solution of the optimization to support rapid turnaround of analyses for high consequence decision making (e.g., public health (Ozik et al., 2021), meteorological (Goubier et al., 2020), and other emergency response (Mandel et al., 2019)). This is sometimes called the high throughput regime (Hernández-Lobato et al., 2017). Hence the time dedicated to select the batch points should be minimal (and not considered negligible), as well as amenable to parallelization (to use the available computing concurrency).

The method we propose is to directly identify candidates realizing different exploration/exploitation trade-offs. This amounts to approximating the GP predictive mean vs. variance Pareto front, which is orders of magnitude faster than optimizing most existing batch infill criteria. In doing so, we shift the paradigm of optimizing (or sampling) acquisition functions over candidate batches to quickly finding a set of desirable candidates to choose from. In the noisy case, possibly with input-dependent variance, the variance reduction makes an additional objective to further encourage replication. Then, to actually select batches, we follow the approach proposed by (Guerreiro and Fonseca, 2016) with the hypervolume Sharpe ratio indicator (HSRI) in the context of evolutionary algorithms. In the MO version, the extension is to take the mean and variance of each objective, and still select batch-candidates based on the HSRI. The contributions of this work are: **1)** The use of a portfolio allocation strategy for batch-BO, defined on the exploration/exploitation trade-off. It extends directly to the multi-objective setup; **2)** An approach independent of the size of the batch, removing limitations of current batch criteria for large $q$; **3)** The potential for flexible batch sizes and asynchronous allocation via the portfolio approach; **4)** The ability to natively take into account replication and to cope with input-dependent noise variance.

In Section 2 we briefly present GPs, batch BO and MO BO as well as their shortcomings for massive batches. In Section 3, the proposed method is described. It is then tested and compared empirically with alternatives in Section 4. A conclusion is given in Section 5.

## 2 Background and related works

We consider the minimization problem of the black-box function $f$: find $\mathbf{x}^* \in \underset{\mathbf{x} \in \mathbb{X} \subseteq \mathbb{R}^d}{\operatorname{argmin}} f(\mathbf{x})$ where $\mathbb{X}$ is typically a hypercube. Among various options for surrogate modeling of $f$, see e.g., Shahriari et al. (2016), GPs are prevalent.

### 2.1 Gaussian process regression

Consider a set of $n \in \mathbb{N}^*$ designs-observations couples $(\mathbf{x}_i, y_i)$ with $y_i = f(\mathbf{x}_i) + \varepsilon_i$, $\varepsilon_i \sim \mathcal{N}(0, \tau(\mathbf{x}_i))$, often obtained with a Latin hypercube sample as design of experiments (DoE). The idea of GP regression, or kriging, is to assume that $f$ follows a multivariate normal distribution, characterized by an arbitrary mean function $m(\mathbf{x})$ and a positive semi-definite covariance kernel function $k : \mathbb{X} \times \mathbb{X} \to \mathbb{R}$. Unless prior information is available to specify a mean function, $m$ is assumed to be zero for simplicity. As for $k$, parameterized families of covariance functions such as Gaussian or Matérn ones are preferred, whose hyperparameters (process variance $\sigma^2$, lengthscales) can be inferred in many ways, such as maximum likelihood estimation.

Conditioned on observations $\mathbf{y} := (y_1, \ldots, y_n)$, zero mean GP predictions at any set of $q$ designs in $\mathbb{X}$, $\mathcal{X}_q : (\mathbf{x}'_1, \ldots, \mathbf{x}'_q)^\top$, are still Gaussian, $Y_n(\mathcal{X}_q)|\mathbf{y} \sim \mathcal{N}(m_n(\mathcal{X}_q), s_n^2(\mathcal{X}_q))$:

$$m_n(\mathcal{X}_q) = \mathbf{k}_n(\mathcal{X}_q)^\top \mathbf{K}_n^{-1} \mathbf{y}_n,$$
$$s_n^2(\mathcal{X}_q) = k(\mathcal{X}_q, \mathcal{X}_q) - \mathbf{k}_n(\mathcal{X}_q)^\top \mathbf{K}_n^{-1} \mathbf{k}_n(\mathcal{X}_q) + \tau(\mathcal{X}_q)$$

where $\mathbf{k}_n(\mathcal{X}_q) = (k(\mathbf{x}_i, \mathbf{x}'_j))_{1 \leq i \leq n, 1 \leq j \leq q}$ and $\mathbf{K}_n = (k(\mathbf{x}_i, \mathbf{x}_j) + \delta_{i=j}\tau(\mathbf{x}_i))_{1 \leq i, j \leq n}$. We refer to (Rasmussen and Williams, 2006; Forrester et al., 2008; Ginsbourger, 2018; Gramacy, 2020) and references therein for additional details on GPs and associated sequential design strategies.

Noise variance, $\tau(\mathbf{x})$, if present, is seldom known and must be estimated. With replication, stochastic kriging (Ankenman et al., 2010) relies on empirical variance estimates. Otherwise, estimation methods have been proposed, building on the Markov chain Monte Carlo method of (Goldberg et al., 1998), as discussed, e.g., by (Binois et al., 2018). Not only is replication beneficial in terms of variance estimation, it also has an impact on the computational speed of using GPs, where the costs scale with the number of unique designs rather than the total number of evaluations.

## 2.2 Batch Bayesian optimization

Starting from the initial DoE to build the starting GP model, (batch-) BO sequentially selects one ($q$) new design(s) to evaluate based on the optimization of an acquisition function that balances exploration and exploitation. The GP model is updated every time new evaluation results are available. The generic batch BO loop is illustrated in Algorithm 1.

---
**Algorithm 1** Pseudo-code for batch BO
---
**Input:** $N_{max}$ (total budget), $q$ (batch size), GP model trained on initial DoE $(\mathbf{x}_i, y_i)_{1 \leq i \leq n}$
1: **while** $n \leq N_{max}$ **do**
2:     Choose $\mathbf{x}_{n+1}, \ldots, \mathbf{x}_{n+q} \in \arg\max_{\mathcal{X}_q \in \mathbb{X}} \alpha(\mathcal{X}_q)$
3:     Update the GP model by conditioning on $\{\mathbf{x}_{n+i}, y_{n+i}\}_{1 \leq i \leq q}$.
4:     $n \leftarrow n + q$
5: **end while**
---

Among acquisition functions $\alpha$, we focus on EI, with its analytical expression, compared with, say, entropy criteria. EI (Mockus et al., 1978) is defined as: $\alpha_{EI}(\mathbf{x}) := \mathbb{E}[\max(0, T - Y_n(\mathbf{x}))]$ where $T$ is the best value observed so far in the deterministic case. In the noisy case, taking $T$ as the best mean estimation over sampled designs (Villemonteix et al., 2009a) or the entire space (Gramacy and Lee, 2011), are alternatives. Integrating out noise uncertainty is done by Letham et al. (2018), losing analytical tractability. This acquisition function can be extended to take into account the addition of $q$ new points, e.g., with the batch (q in short) EI, $\alpha_{qEI}(\mathcal{X}_q) := \mathbb{E}[\max(0, T - Y_n(\mathbf{x}'_1), \ldots, T - Y_n(\mathbf{x}'_q)]$ that has an expression amenable for computation (Chevalier and Ginsbourger, 2013) and also for its gradient (Marmin et al., 2015). A much faster approximation of the batch EI (qEI) is described in Binois (2015), relying on nested Gaussian approximations of the maximum of two Gaussian variables from Clark (1961). Otherwise, stochastic approximation (Wang et al., 2020) or sampling methods by Monte Carlo, e.g., with the reparameterization trick (Wilson et al., 2018), are largely used, but may be less precise as the batch size increases.

Many batch versions of infill criteria have been proposed, such as (Kandasamy et al., 2018; Hernández-Lobato et al., 2017) for Thompson sampling. For EI, rather than just looking at its local optima (Sóbester et al., 2004), some heuristics propose to select batch points iteratively, replacing unknown values at selected points by pseudo-values (Ginsbourger et al., 2010). This was coined as "hallucination" in the UCB version of (Desautels et al., 2014). More generally, Rontsis et al. (2020) use an optimistic bound on EI for all possible distributions compatible with the same first two moments as a GP, which requires solving a semi-definite problem, limiting the scaling up to large batches. Gonzalez et al. (2016) reduce batch selection cost by not modeling the joint probability distribution of the batch nor using a hallucination scheme. Their idea is to select batch members sequentially by penalizing proximity to the previously selected ones. Taking different infill criteria is an option to select different trade-offs, as by (Tran et al., 2019). This idea of a portfolio of

acquisition functions is also present in (Hoffman et al., 2011), but limited to a few options and not intended as a mechanism to select batch candidates. Using local models is another way to select batches efficiently, up to several hundreds in (Wang et al., 2018). The downside is a lack of coordination in the selection and the need of an *ad hoc* selection procedure. For entropy or stepwise uncertainty reduction criteria, see, e.g., (Chevalier et al., 2014a), batching would increase their already intensive computational burden. Another early work by (Azimi et al., 2010) attempts to match the expected sequential performance, via approximations and sampling.

## 2.3   Multi-objective BO

The multi-objective optimization problem (MOOP) is to find $\mathbf{x}^* \in \underset{\mathbf{x} \in \mathbb{X} \subset \mathbb{R}^d}{\operatorname{argmin}}(f_1(\mathbf{x}), \dots, f_p(\mathbf{x}))$, $p \geq 2$. $p > 4$ is often called the *many-objective* setup, with its own set of challenges for BO, see e.g., (Binois et al., 2020). The solutions of a MOOP are the best compromise solutions between objectives, in the Pareto dominance sense. A solution $\mathbf{x}$ is said to be dominated by another $\mathbf{x}'$, denoted $\mathbf{x}' \preceq \mathbf{x}$, if $\forall i, f_i(\mathbf{x}') \leq f_i(\mathbf{x})$ and $f_i(\mathbf{x}') < f_i(\mathbf{x})$ for at least one $i$. The Pareto set is the set of solutions that are not dominated by any other design in $\mathbb{X}$: $\left\{ \mathbf{x} \in \mathbb{R}^d, \nexists \mathbf{x}' \in \mathbb{R}^d \text{ such that } \mathbf{x}' \preceq \mathbf{x} \right\}$; the Pareto front is its image in the objective space. In the noisy case, we consider the noisy MOOP defined on expectations over objectives (Hunter et al., 2017).

Measured in the objective space, the hypervolume refers to the volume dominated by a set of points relative to a reference point, see Figure 1. It serves both as a performance metric in MOO, see e.g., Audet et al. (2020) and references therein, or to measure improvement in extending EI (Emmerich et al., 2006). The corresponding expected hypervolume improvement (EHI) can be computed in closed form for two or three objectives (Emmerich et al., 2011; Yang et al., 2017), or by sampling, see e.g., Svenson (2011). A batch version of EHI, qEHI, is proposed by Daulton et al. (2020; 2021). The operations research community has seen works dealing with low signal-to-noise ratios and heteroscedasticity, where replication is key. Generally, the idea is to first identify good points before defining the number of replicates, see, e.g., (Zhang et al., 2017; Gonzalez et al., 2020) or (Rojas-Gonzalez and Van Nieuwenhuyse, 2020) for a review on stochastic MO BO. Still, the batch aspect is missing in the identification of candidates.

## 2.4   Challenges with massive batches

There are several limitations in the works above. First, while it is generally assumed that the cost of one evaluation is sufficiently high to consider the time to select new points negligible, this may not be the case in the large batch setup. Parallel infill criteria are more expensive to evaluate, and even computational costs increasing linearly in the batch size ($q$) become impractical for hundreds or thousands of batch points. For instance, the exact qEI expression uses multivariate normal probabilities, whose computation do not scale well with the batch size. There are also many approximated criteria for batch EI, or similar criteria. However, in all current approaches, the evaluation costs increase with the batch size, at best linearly in $q$ for existing criteria, which remains too costly for the regime we target.

This is already troublesome when optimization iterations must be conducted quickly, but is amplified by the difficulty of optimizing the acquisition function itself. While earlier works used branch and bounds (Jones et al., 1998) to guarantee optimality with $q = 1$, multi-start gradient based optimization or EAs are predominantly used. In the batch setting, the size of the optimization problem becomes $q \times d$, a real challenge, even with the availability of gradients. Wilson et al. (2018) showed that greedily optimizing batch members one-by-one is sensible, which still requires to solve $q$ $d$-dimensional global optimization problems. Both become cumbersome for large batches and, presumably, far from the global optimum since the acquisition function landscape is multimodal, with flat regions, and symmetry properties. Plus, as we showcase, this results in parts of batch members being less pertinent.

Relying on discrete search spaces bypasses parts of the issue, even though finding the best batch becomes a combinatorial search. In between the greedy and joint options is the work by Daxberger and Low (2017), to optimize an approximated batch-UCB criterion as a distributed constraint problem. As a result, only sub-optimal solutions are reachable in practice for batch acquisition function optimization. Rather than optimizing, a perhaps even more computationally intensive option is to consider the density under EI. That

is, to either find local optima and adapt batch size as in Nguyen et al. (2016), or sampling uniformly from the EI density with slice sampling and clustering as with Groves and Pyzer-Knapp (2018). Thompson sampling for mono-objective batch BO, see e.g., Kandasamy et al. (2018), also bypasses the issues of optimizing the acquisition function, but batch members are independently obtained, which can be wasteful for large batches. Lastly, adaptive batch sizes might be more efficient than a fixed number of parallel evaluations, see e.g., (Desautels et al., 2014). Similarly, asynchronous evaluation is another angle to exploit when the simulation evaluation times vary, see e.g., (Gramacy and Lee, 2009; Janusevskis et al., 2012; Alvi et al., 2019).

Replication adds another layer of decision: whether or not adding a new design is worth the future extra computational cost, compared to the perhaps marginally worse option of replicating on the acquisition function value. With high noise, choosing the amount of replication becomes important, as individual evaluations contain almost no information. But even fixing the number of replicates per batch, selecting batch design locations plus replication degree makes a hard dynamic programming optimization problem.

## 3 Batch selection as a portfolio problem

We propose an alternative to current BO methods to handle large batches by returning to the roots of BO, with the exploration/exploitation trade-off. Specifically, we focus on a portfolio selection criterion to select a batch balancing risk and return.

### 3.1 Exploration/exploitation trade-off

At the core of BO is the idea that regions of interest have either a low mean, or have a large variance. This is the BO exploration/exploitation trade-off, see e.g., Garnett (2022). From a multi-objective point of view, acquisition functions resolve this trade-off by selecting a solution on the corresponding mean vs. standard deviation $(m_n/s_n)$ Pareto front $\mathcal{P}$. With UCB (Srinivas et al., 2010), $\alpha_{UCB}(\mathbf{x}) := m_n(\mathbf{x}) - \sqrt{\beta} s_n(\mathbf{x})$, the tuning parameter $\beta$ is a way to select one solution on the convex parts of this Pareto front. EI can be interpreted this way as well, as noticed by Jones et al. (1998); De Ath et al. (2021) in showing that $\frac{\partial EI}{\partial m_n}(x) = -\Phi\left(\frac{T-m_n(\mathbf{x})}{s_n(\mathbf{x})}\right) < 0$ and $\frac{\partial EI}{\partial s_n}(x) = \phi\left(\frac{T-m_n(\mathbf{x})}{s_n(\mathbf{x})}\right) > 0$, where $\phi$ (resp. $\Phi$) are the Gaussian pdf (resp. cdf). Hence EI also selects a specific solution on the corresponding Pareto front. Navigating the Pareto front can be done by taking expectations of powers of the improvement, i.e., the generalized EI (GEI) (Schonlau et al., 1998; Wang et al., 2017), for which higher powers of EI reward larger variance and make it more global, as in Ponweiser et al. (2008). Note that the probability of improvement (PI), $\alpha_{PI} = \mathbb{P}(Y_n(\mathbf{x}) < T)$, which corresponds to the zeroth-order EI, is not on the trade-off Pareto front $\mathcal{P}$, explaining why PI is often discarded as being too exploitative (higher variance is detrimental as soon as the predicted mean is below $T$). Our main point is that rather than having to define a specific trade-off between exploration and exploitation *a priori*, before considering batching, it is better to find the set of optimal trade-offs $\mathcal{P}$ and select batch points from it *a posteriori*.

### 3.2 Portfolio selection with HSRI

We propose to use a criterion to select solutions on the exploration-exploitation Pareto front, rather than doing so randomly as in Gupta et al. (2018). Yevseyeva et al. (2014) defined the hypervolume Sharpe ratio indicator (HSRI) to select individuals in MO EAs, with further study on their properties in Guerreiro and Fonseca (2016; 2020). Here, we borrow from this portfolio-selection approach, where individual performance is related to expected return, while diversity is related to the return covariance.

Let $A = \{\mathbf{a}^{(1)}, \dots, \mathbf{a}^{(l)}\}$ be a non-empty set of assets, $\mathbf{a}^{(i)} \in \mathbb{R}^s$, $s \geq 2$, let vector $\mathbf{r} \in \mathbb{R}^l$ denote their expected return and matrix $\mathbf{Q} \in \mathbb{R}^{l \times l}$ denote the return covariance between pairs of assets. Let $\mathbf{z} \in [0,1]^l$ be the investment vector where $z_i$ denotes the investment in asset $\mathbf{a}^{(i)}$. The Sharpe-ratio maximization problem is defined as

$$\max_{\mathbf{z} \in [0,1]^l} h(\mathbf{z}) = \frac{\mathbf{r}^\top \mathbf{z} - r_f}{\sqrt{\mathbf{z}^\top \mathbf{Q} \mathbf{z}}} \ s.t. \ \sum_{i=1}^{l} z_i = 1, \tag{1}$$

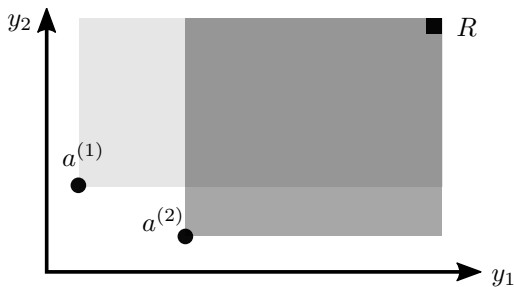

Figure 1: Hypervolume dominated by two assets $a^{(1)}$ (light gray) and $a^{(2)}$ (gray) with respect to the reference point $R$, corresponding to the expected return. The covariance return is given by the volume jointly dominated by both points (dark gray).

with $r_f$ the return of a riskless asset and $h$ the Sharpe ratio. This problem, restated as a convex quadratic programming problem (QP), see e.g., Guerreiro and Fonseca (2016), can be solved efficiently only once per iteration. The outcome is a set of weights, corresponding to the allocation to each asset.

HSRI Guerreiro and Fonseca (2016) is an instance of portfolio selection where the expected return and return covariance are based on the hypervolume improvement: $r_i = p_{ii}$ and $Q_{ij} = p_{ij} - p_{ii}p_{jj}$ where $p_{ij} = \left( \prod_{1 \leq t \leq p} (R_l - \max(a_t^{(i)}, a_t^{(j)})) \right) / \left( \prod_{1 \leq t \leq p} (R_l - f_l^*) \right)$; see Figure 1 for an illustration. Note that this hypervolume computation scales linearly with the number of objectives. Importantly, as shown in Guerreiro and Fonseca (2016), if a set of assets is dominated by another set, its Sharpe ratio is lower. Furthermore, no allocation is made on dominated points: they are all on $\mathcal{P}$. Finally they show that only the reference point $R$ needs to be set in practice.

### 3.3 Proposition with qHSRI

From a BO viewpoint, the goal is to obtain the $q$ candidates leading to the highest return in terms of HSRI: $\alpha_{qHSRI}(\mathcal{X}_q) = h(z(\mathcal{X}_q)) = \left( \mathbf{r}(\mathcal{X}_q)^\top \mathbf{1}_q - r_f \right) \left( \sqrt{\mathbf{1}_q^\top \mathbf{Q}(\mathcal{X}_q) \mathbf{1}_q} \right)^{-1}$ with $\mathbf{1}_q$ a vector of $q$ ones. Here, instead of using actual objective values as in MO EAs, an asset $\mathbf{a}^{(i)}$, corresponding to candidate design $\mathbf{x}^{(i)}$, is characterized by its GP predictive mean(s) and standard deviation(s), i.e., with $p = 1$, $a_1^{(i)} = m_n(\mathbf{x}_i)$ and $a_2^{(i)} = -s_n(\mathbf{x}^{(i)})$. Also here $r_f = 0$ since risk-less assets (noiseless observations) bring no improvement. Since the optimal solutions will belong to $\mathcal{P}$ thanks to the properties of HSRI, the search can be decomposed into three steps: approximating $\mathcal{P}$, solving the QP problem (1) over this approximation of $\mathcal{P}$, and then selecting $q$ evaluations. In the noiseless case, they are the $q$ largest weights. When replication is possible, the allocation becomes proportional to the weights: find $\gamma$ s.t. $\sum_{i=1}^{l} \lfloor \gamma \times z_i^* \rceil = q$, by bisection (randomly resolving ties). The adaptation to the asynchronous setup is detailed in Appendix A.

**Extension to the multi-objective setup** To extend to MO, we propose to search trade-offs on the objective means, $m_n^{(1)}(\mathbf{x}), \ldots, m_n^{(p)}(\mathbf{x})$, and averaged predictive standard deviations Pareto front, $\bar{\sigma}_n(\mathbf{x}) = p^{-1} \sum_{i=1}^{p} s_n^{(i)}(\mathbf{x})/\sigma_n^{(i)}$ with $(\sigma_n^{(i)})^2$ the $i^{th}$-objective GP variance hyperparameter, a $p + 1$ dimensional space. Taking all $p$ standard deviations is possible, but the corresponding objectives are correlated since they increase with the distance to observed design points. In the case where the GP hyperparameters are the same and evaluations of objectives coupled, the objectives would be perfectly correlated. Even with different hyperparameters, preliminary tests showed that using the averaged predictive standard deviation do not degrade the performance compared to the additional difficulty of handling more objectives.

**Replication** When noise is present, we include an additional objective of variance reduction. That is, for two designs with the same mean and variance, the one for which adding an observation will decrease the predictive variance the most is preferred. This decrease is given by GP update equations, see e.g.,

Chevalier et al. (2014b), and does not depend on the value at the future $q$ designs: $s^2_{n+q}(\mathcal{X}_q) = s^2_n(\mathcal{X}_q) - s^2_n(\mathcal{X}_q, \mathbf{x}_{1:(n+q)})(s^2_n(\mathbf{x}_{1:(n+q)}))^{-1}s^2_n(\mathbf{x}_{1:(n+q)}, \mathcal{X}_q)$, with $\mathbf{x}_{1:(n+q)}$ defined as the current DoE augmented by future $q$ designs. It does depend on the noise variance and the degree of replication, see, e.g., Binois et al. (2019), which may be used to define a minimal degree of replication at candidate designs to ensure a sufficient decrease. Similarly, it is possible to limit the replication degree when the decrease of variance of further replication is too low.

The pseudo-code of the approach is given in Algorithm 2. The first step is to identify $s$ designs on the mean vs. standard deviation Pareto set. In the deterministic case $s$ must be at least equal to $q$, while with noise and replication it can be lower. Population based evolutionary algorithms can be used here, with a sufficiently large population. Once these $s$ candidates are identified, dominated points can be filtered out as HSRI only selects non-dominated solutions. Points with low probability of improvement (or with low probability of being non-dominated) can be removed as well. In the noisy case, the variance reduction serves as an additional objective for the computation of HSRI. It is integrated afterwards as it is not directly related to the identification of the exploration-exploitation tradeoff surface. Computing qHSRI then involves computing $\mathbf{r}$ and $\mathbf{Q}$ before solving the corresponding QP problem (1). Designs from $\mathbf{X}_s$ are selected based on $\mathbf{z}^*$: either by taking the $q$ designs having the largest weights $z_i$, or computing $\gamma$ to obtain an allocation, which can include replicates.

---

**Algorithm 2** Pseudo-code for batch BO with qHSRI

---

**Input:** $N_{max}$ (total budget), $q$ (batch size), $p$ GP model(s) fitted on initial DoE $(\mathbf{x}_i, y_i)_{1 \leq i \leq n}$

1: **while** $n \leq N_{max}$ **do**
2:     Find $\mathbb{X}_s \in \arg\min(m_n^{(1)}(\mathbf{x}), \ldots, m_n^{(p)}(\mathbf{x}), \bar{\sigma}_n(\mathbf{x}))$
3:     Filter dominated solutions in $\mathbb{X}_s$
4:     **if** $p = 1$ **then**
5:         Filter points with low PI in $\mathbb{X}_s$
6:     **else**
7:         Filter points with low PND in $\mathbb{X}_s$
8:     **end**
9:     If $\tau(\mathbf{x}) > 0$: add variance reduction to objectives
10:     Compute return $\mathbf{r}$ and covariance matrix $\mathbf{Q}$
11:     Compute optimal Sharpe ratio $\mathbf{z}^*$
12:     Allocate $q$ points based on the weights
13:     Update the GP model(s) with $\{\mathbf{x}_{n+i}, y_{n+i}\}_{1 \leq i \leq q}$.
14:     $n \leftarrow n + q$
15: **end while**

---

In terms of complexity, the main task is to find the assets, i.e., candidate designs on $\mathcal{P}$. Evaluating $m_n$ and $s_n$ cost $\mathcal{O}(n^2)$ after a $\mathcal{O}(n^3)$ matrix inversion operation that only needs to be done once. An order of magnitude can be gained with approximations, see for instance Wang et al. (2019). Then $\mathbf{r}$ and $\mathbf{Q}$ are computed on the assets, before maximizing the Sharpe ratio, whose optimal weights provide the best $q$ designs. Filtering solutions reduces the size of the QP problem to be solved, either with PI or the probability of non-domination (PND) in the MO case. Crucially, the complexity does not depend on $q$ with replication.

We illustrate the proposed method (qHSRI) on an example in Figure 2, comparing the outcome of optimizing qEI, its fast approximation qAEI (Binois, 2015) and qHSRI in the mean vs. standard deviation ($m_n$ vs. $s_n$) space. Additional candidates are shown, either randomly sampled in $\mathbb{X}$ or on the estimated exploration/exploitation Pareto set. Negation of the standard deviation is taken for minimization. While all qHSRI selected designs are on $\mathcal{P}$, this is not the case for the qEI version, particularly so when $q$ is larger, where none of the selected designs are – possibly due to the much higher difficulty of solving the corresponding optimization problem. Designs corresponding to powers of EI also appear on $\mathcal{P}$, showing a richer exploration/exploitation trade-off than with EI only. We also observe that points with large variance may not be of interest if they have a negligible PI (e.g., $< 0.1$).

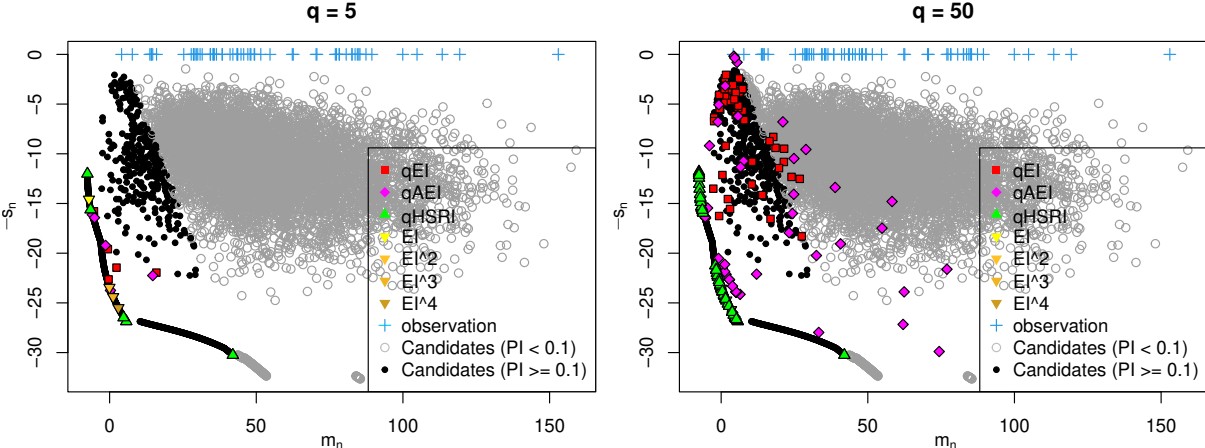

Figure 2: Comparison of qHSRI with qEI and qAEI acquisition functions on the noiseless repeated Branin function ($d = 6$, $n = 60$). The first four GEI optimal solutions are depicted as well.

It could be argued that the qHSRI approach ignores distances in the input space and could form clusters. While this is the case, depending on the exploration-exploitation Pareto set, since the batch points cover $\mathcal{P}$, it automatically adapts to this latter's range of optimal values, depending on the iteration and problem. This is harder to adapt *a priori* in the input space and it avoids having to define minimal distances manually, as in Gonzalez et al. (2016). Still, for numerical stability and to favor replication, a minimal distance can be set as well.

## 4 Experiments

Except Wang et al. (2018) that uses disconnected local GP models and Gupta et al. (2018) that also uses the exploration-exploitation PF, existing batch BO methods mostly give results with low $q$, e.g., $q \leq 20$. On the implementations we could test, these criteria take more than seconds per evaluation with $q \approx 100$, while, in our approach, predicting for a given design takes less than a millisecond. Consequently, comparisons with qHSRI are not feasible for massive $q$. We provide scaling results for larger $q$ values with qHSRI in Appendix B.

The `R` package `hetGP` (Binois and Gramacy, 2021) is used for noisy GP modeling. Anisotropic Matérn covariance kernels are used throughout the examples, whose hyperparameters are estimated by maximum likelihood. As we use the same GP model, the comparison shows the effect of the acquisition function choice: qEI or qEHI vs. qHSRI. qEI is either the exact version (Chevalier and Ginsbourger, 2013) in `DiceOptim` (Picheny et al., 2021), or the approximated one from Binois (2015), qAEI. qEI is not available for $q > 20$ nor in the noisy case. In the mono-objective case, we also include Thompson sampling, qTS, implemented by generating GP realisations on discrete sets of size $200d$. qEHI is from `GPareto` (Binois and Picheny, 2019), estimated by Monte Carlo. PF denotes the method from Gupta et al. (2018), where the batch members are randomly selected on the exploration-exploitation Pareto front. Random search (RS) is added as a baseline. All start with the same space filling designs of size $5d$, replicated five times each to help the heteroscedastic GP modeling with low signal to noise ratios.

Optimization of the acquisition functions is performed by combining random search, local optimization and EAs. That is, for qEI, $n_u = 100d$ designs are uniformly sampled in the design space before computing their univariate EI. Then $n_b = 100d$ candidate batches are created by sampling these designs with weights given by EI. Then the corresponding best batch for qEI is optimized locally with L-BFGS-B. A global search with `pso` (Bendtsen, 2012) (population of size 200) is conducted too, to directly optimize qEI, and the overall best qEI batch is selected. The same principle is applied for qEHI. As for qHSRI and PF, in addition to the same uniform sampling strategy with $n_u$ designs, NSGA-II (Deb et al., 2002) from `mco` (Mersmann, 2020) is used to find $\mathcal{P}$, the exploration/exploitation compromise surface (with a population of size 500). The reference

point $R$ for HSRI computations is obtained from the range over each component, extended by 20%, as is the default in `GPareto` Binois and Picheny (2019). The `R` code (R Core Team, 2023) of the approach is available as supplementary material.

For one objective, the optimality gap, i.e., the difference to a reference solution, is monitored. With noise, the optimality gap is computed both on noiseless values (when known) and on the estimated minimum by each method over iterations, which is the only element accessible in real applications. In fact, the optimality gap only assesses whether a design has been sampled close to an optimal one, not if it has been correctly predicted. The hypervolume metric is used in the MO case, from a reference Pareto front computed using NSGA-II and all designs found by the different methods. Similar to the mono-objective case, the hypervolume difference is also computed on the estimated Pareto set by each method, over iterations, to have a more reliable and realistic performance monitoring.

In a first step, for validating qHSRI, we compared it with alternatives for relatively low $q$ values. These preliminary experiments on noiseless functions are provided in Appendix B. The outcome is that qHSRI gives results on par with qEI and qEHI looking at the performance over iterations, at a much lower computational cost. These results also motivate the use of qAEI as a proxy for qEI when this latter is not available. We notice that qTS performed poorly on these experiments, possibly because using discretized realisations is insufficient for the relatively large input dimension ($d = 12$). There the use of random or Fourier features may help Mutny and Krause (2018). As for PF, it requires more samples than qHSRI, even if it goes as fast. This indicates that the portfolio allocation strategy is beneficial compared to randomly sampling on the exploration-exploitation Pareto front.

## 4.1 Mono-objective results

We first consider the standard Branin and Hartmann6 test functions, see e.g., (Roustant et al., 2012). For the noisy Branin (resp. Hartmann6), we take the first objective of the P1 test function (Parr, 2012) (resp. repeated Hartmann3 (Roustant et al., 2012)) as input standard deviation $\tau(\mathbf{x})^{\frac{1}{2}}$, hence with heteroscedastic noise (denoted by *het* below).

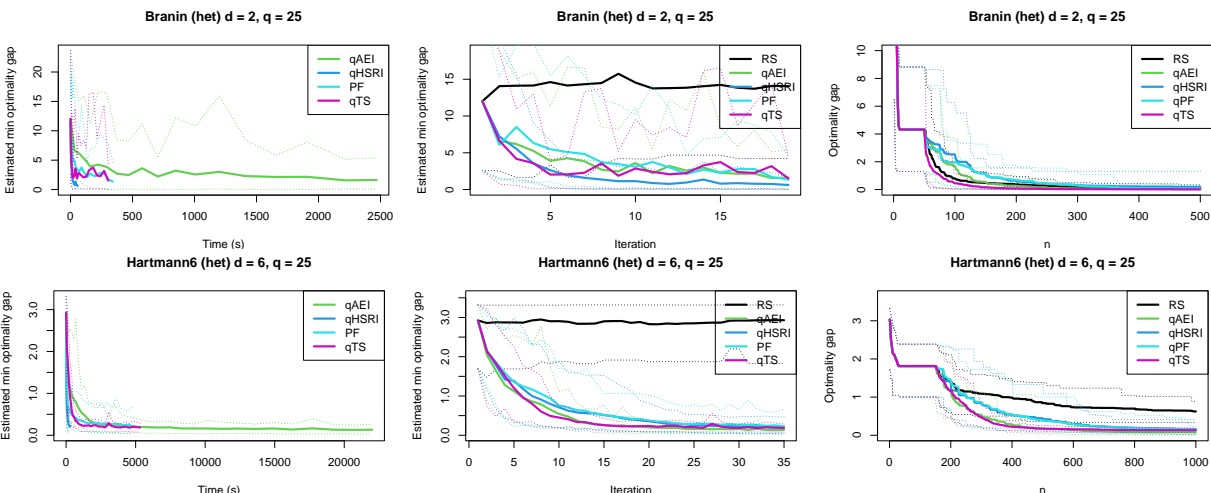

Figure 3: Mono-objective results over iterations and over time. Optimality gap and estimated optimality gap for noisy tests over 40 macro-runs are given. Thin dashed lines are 5% and 95% quantiles.

Figure 3 highlights that qHSRI is orders of magnitude faster than qTS and qAEI for decreasing the optimality gap (see also Table 1 for all timing results). It also improves over random selection on $\mathcal{P}$ as with PF. In these noisy problems, looking at the true optimality gap for observed designs shows good performance of RS, since, especially in small dimension like for Branin ($d = 2$), there is a high probability of sampling close to one of its three global minima. Also, replicating incurs less exploration, penalizing qHSRI on this metric. Nevertheless, the actual metric of interest is the optimality gap of the estimated best solution at each iteration. It is

improved with qHSRI, especially for Branin, while performances of the various acquisition functions are similar iteration-wise in the Hartmann6 case, with qTS being slightly better initially. Concerning speed, part of the speed-ups are due to the ability to replicate. Indeed, as apparent in supplementary Figure 9, for qHSRI the number of unique designs remains low, less than 20% of the total number of observations without degrading the sample efficiency. Notice that taking larger batches can be faster since the batch selection is independent of $q$ with qHSRI. Also there are fewer iterations for the same simulation budget, hence less time is spent in fitting GPs and finding $\mathcal{P}$.

Next we tackle the more realistic $12d$ Lunar lander problem (Eriksson et al., 2019). We take $N_{max} = 2000$ with $q = 100$, where a single evaluation is taken as the average over 100 runs to cope with the low signal-to-noise ratio (rather than fixing 50 seeds to make it deterministic as in Eriksson et al. (2019)). The solution found (predicted by the GP) is of $-205.32$ while the reference handcrafted solution gives $-101.13$, see Figure 5. The Lunar lander problem with qHSRI took 5 hours; it did not complete with qAEI even in 24 hours due to $q = 100$.

## 4.2 Multi-objective results

We consider the P1 (Parr, 2012) and P2 (Poloni et al., 2000) problems. For the noisy setup, one problem serves as the other one's noise standard deviation function (taking absolute values for positiveness). The results are shown in Figure 4, where the leftmost panels show the beneficial performance of the qHSRI approach in terms of time to solution. While RS performs relatively well looking solely at the hypervolume difference for the evaluated designs (rightmost panels), this does not relate to the quality of the Pareto front estimation. Indeed, the estimated Pareto with RS, that is, using only the non-dominated noisy observations, is far from the actual Pareto front, due to noise realizations. There the Pareto fronts estimated by GP models do show a convergence to the reference Pareto front, indicating that their estimation improves over iterations (middle panels). Finally, like in the mono-objective results, we demonstrate in Appendix B that for the noiseless case the sample efficiency of qHSRI is at least on par to that of qEHI, and even slightly better in some cases.

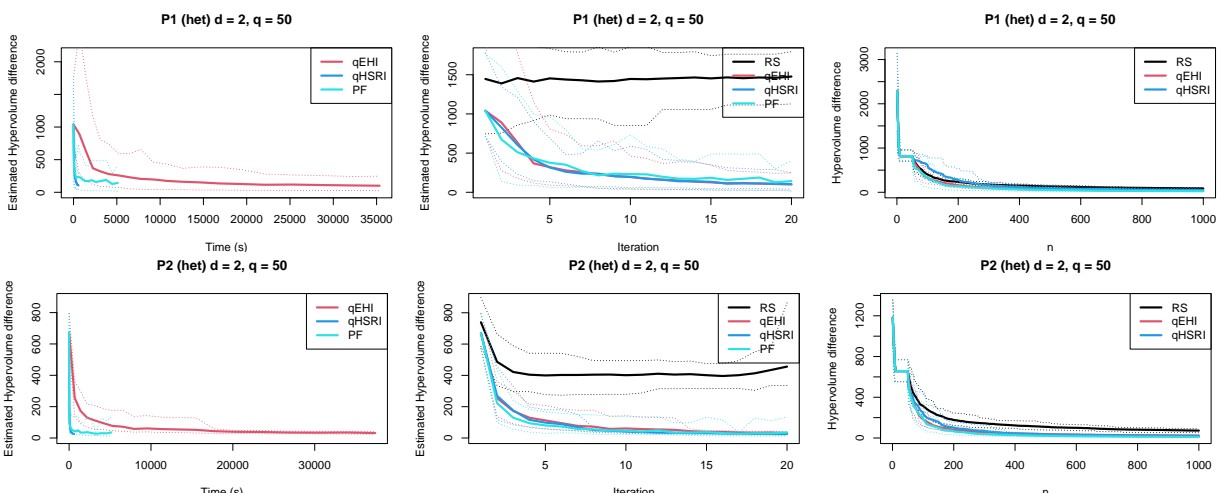

Figure 4: Multi-objective results over time and iterations. Hypervolume difference over a reference Pareto front and its counterpart for the estimated Pareto set for noisy tests over 40 macro-runs are given. Thin dashed lines are 5% and 95% quantiles.

## 4.3 CityCOVID data set

We showcase a motivating application example for massive batching: calibrating the CityCOVID ABM of the population of Chicago in the context of the COVID-19 pandemic, presented in Ozik et al. (2021) and built on the ChiSIM framework (Macal et al., 2018). It models the 2.7 million residents of Chicago as

they move between 1.2 million places based on their hourly activity schedules. The synthetic population of agents extends an existing general-purpose synthetic population Cajka et al. (2010) and statistically matches Chicago's demographic composition. Agents colocate in geolocated places, which include households, schools, workplaces, etc. The agent hourly activity schedules are derived from the American Time Use Survey and the Panel Study of Income Dynamics and assigned based on agent demographic characteristics. CityCOVID includes COVID-19 disease progression within each agent, including differing symptom severities, hospitalizations, and age-dependent probabilities of transitions between disease stages.

The problem is formulated as a nine variable bi-objective optimization problem: the aggregated difference for two target quantities. It corresponds to the calibration of the CityCOVID parameters $\boldsymbol{\theta}$ listed in Table 3, each normalized to $[0, 1]$. Model outputs are compared against two empirical data sources obtained through the City of Chicago data portal City of Chicago (2022): $\mathbf{H}$ the daily census of hospital beds occupied by COVID-19 patients and $\mathbf{D}$ the COVID-19 attributed death data in and out of hospitals, both for residents of Chicago. We used an exponentially weighted error function $L(\boldsymbol{\theta}, T_i, \tilde{T}_i, d), i \in \{\mathbf{H}, \mathbf{D}\}$, with daily discount rate $d$ tuned to 98% and 95% for $\mathbf{H}$ and $\mathbf{D}$, with the corresponding observed (resp. simulated) time-series denoted by $T$ and $\tilde{T}$ giving the objectives.

To inform public health policy, many simulations are necessary in a short period of time, which can only be achieved by running many concurrently. One simulation takes $\approx$ 10min, with a low signal-to-noise ratio. A data set of $217,078$ simulations (over $8,368$ unique designs, with a degree of replication between 1 and 1000) has been collected by various strategies: IMABC (Rutter et al., 2019), qEHI with fixed degree of replication, and replicating non-dominated solutions. This data set is available in the supplementary material.

Contrary to the previous test cases that were defined over continuous domains, for testing qHSRI we use this existing data set. The initial design is a random subset of the data: $25,000$ simulation results over $2500$ unique designs with a degree of replication of $10$, out of $50,585$ simulations over $5075$ unique designs, with a degree of replication between 3 and 10. These correspond to results given by IMABC (akin to a non uniform sampling). qHSRI is used to select candidates among remaining designs up to $q = 2500$ if enough replicates are available, hence with a flexible batch size. To speed up prediction and to benefit from a parallel architecture, local GPs are built from 20 nearest neighbors rather than relying on a single global GP. We show in Figure 5 the progress in terms of symmetric difference to the final estimate of the Pareto front, thus penalizing both under and over confident predictions. qHSRI quickly converges to the reference Pareto front, compared to RS.

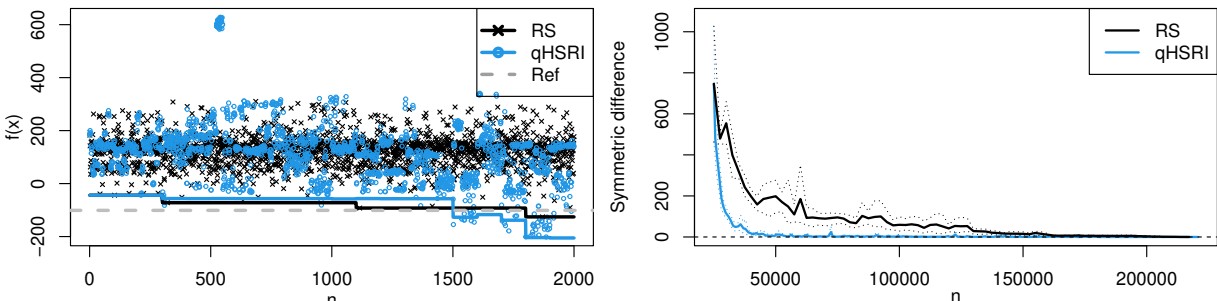

Figure 5: Left: optimality gap for the Lunar lander problem (one single run) with the evaluated values and estimated minimum found. Right: results on CityCOVID data set over 5 macro-runs, thin dashed lines are 5% and 95% quantiles.

## 5 Conclusions and perspectives

Massive batching for BO comes with the additional challenge that batch selection must happen at a fast pace. Here we demonstrate qHSRI as a flexible and light-weight option, independent of the batch size. It also handles replication natively, resulting in additional speed-up for noisy simulators without fixing a degree

of replication. Hence, this proposed approach makes a sensible, simple, yet efficient, baseline for massive batching.

Possible extensions could take into account global effects (e.g., on entropy, integrated variance, etc.) of candidate designs to be less myopic. A more dynamic Sharpe ratio allocation could be beneficial, to improve replication. Finally, while the integration of a few constraints is straightforward, handling more could rely on the use of copulas as in Eriksson and Poloczek (2021) to alleviate the increase of the dimension of the exploration/exploitation trade-off surface. The study of the convergence, e.g., based on results for UCB with various $\beta$s, is left for future work.

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

# A   Adaptation to Asynchronous Batch Bayesian optimization

The standard batch- (or multi-points) BO loop, where parallel evaluations are conducted in parallel and waiting for all of them to finish, is presented in Algorithm 1. Next we discuss briefly the adaptations required for the asynchronous setup.

The main feature of the portfolio approach is to give a weight vector corresponding to Pareto optimal points on the mean/standard deviation Pareto front $\mathcal{P}$. In the synchronous setting, $q$ points are selected based on the weights. That is, the $q$ best ones in the noiseless setting while replicates can occur in the noisy setting. If $q'$ additional points can be evaluated, without new data becoming available (or while waiting for new candidates to evaluate), the change in the asynchronous setup amounts to considering that $q + q'$ batch points (assets) can be selected according to the weights. The $q$ first ones will remain unchanged while $q'$ additional ones can be run. An example is provided in Figure 6.

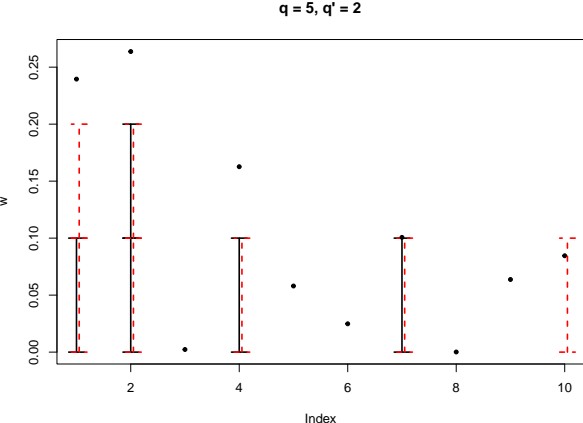

Figure 6: Batch allocation on indices according to weights $w$ (black points) for 10 candidates. $q = 5$ designs are initially allocated, as represented by the black segments. If $q' = 2$ additional designs can be evaluated, then the allocation is recomputed with the same weights and leads to the red dashed segments.

# B   Additional experimental results

We complement the results of Section 4 on noisy problems with results on noiseless ones. The R package `DiceKriging` (Roustant et al., 2012) is used for deterministic GP modeling. In this deterministic setup, $d$ is increased to accommodate larger batches via repeated versions of these problem as used, e.g., in Oh et al. (2018).

The timing results of all synthetic benchmarks are presented in Tables 1 and 2, with best results in bold. qHSRI or PF are always much faster than the alternatives, even the approximated qEI criterion (qAEI) or qTS. The R code (R Core Team, 2023) of the approach and the CityCOVID data are available as supplementary material. Results have been obtained in parallel on dual-Xeon Skylake SP Silver 4114 @ 2.20GHz (20 cores) and 192 GB RAM (or similar nodes). Lunar lander tests have been run on a laptop.

The detailed results of the performance over time are given in Figure 7, where qHSRI show better results than the alternatives, closely matched by PF except on Branin where it does worse. qTS performs badly as the discretization is insufficient given the problem dimension.

The results over iterations of Figure 8 shows that the performance of qHSRI matches the one of qEI and improves over qEHI. qTS underperforms but it is still better than RS. PF shows similar results to qHSRI on the multi-objective problems and Hartmann6 with $q = 25$, but is worse in the other test cases. For the smallest batch size, $q = 10$, qAEI matches the performance of qEI, while being faster.

| f-d-q | qEI | qAEI | qHSRI | qPF | qTS |
|-------|-----|------|-------|-----|-----|
| B-12-10 | 24725 (133) | 5901 (54) | 1103 (104) | **1024** (160) | 1835 (5) |
| B-12-25 | − | 5856 (61) | 451 (46) | **423** (66) | 1760 (3) |
| H-12-10 | 24258 (115) | 4957 (131) | **969** (38) | 1266 (144) | 1938 (22) |
| H-12-25 | − | 4585 (129) | **419** (24) | 510 (76) | 1816 (9) |
| B(h)-2-25 | − | 2470 (93) | **57** (3) | 342 (36) | 305 (46) |
| H(h)-6-25 | − | 22012 (717) | **288** (33) | 4714 (702) | 5327 (503) |

Table 1: Averaged timing results (in s), with standard deviation in parenthesis. B is for Branin, H for Hartmann6, (h) for heteroscedastically noisy problems and '−' indicates when non applicable.

| f-d-q | qEHI | qHSRI | qPF |
|-------|------|-------|-----|
| P1-6-50 | 16308 (2118) | 678 (25) | **580** (26) |
| P1(h)-2-50 | 35350 (1363) | **567** (59) | 5110 (407) |
| P2-6-50 | 14022 (1818) | 635 (24) | **553** (20) |
| P2(h)-2-50 | 37386 (1123) | **608** (44) | 5148 (414) |

Table 2: Averaged timing results (in s), with standard deviation in parenthesis. (h) is for heteroscedastically noisy problems.

To highlight that replication occurs, Figure 9 represents the number of unique design over time for the mono-objective noisy problems. Notice that replication is present in the initial design and that it can happen for all methods when sampling vertices of the hypercube.

We complement the synthetic experiments in Section 4 with results for larger values of $q$ using qHSRI. The mono-objective results are in Figure 10 and the multi-objective ones in 11. Increasing $q$ shows degrading results sample-wise, as there are fewer updates of the GP model. But increasing $q$ decreases the time to solution.

## C  Details on CityCOVID calibration parameters

The nine variables of the simulator are given in Table 3.

| $\theta$ | $\pi(\theta)$ | Description |
|----------|---------------|-------------|
| $\theta_1$ | $U(60, 190)$ | Initial number of exposed (infected but not infectious) agents at the beginning of the simulation |
| $\theta_2$ | $U(0.03, 0.1)$ | Base hourly probability of transmission between one infectious and one susceptible person occupying the same location |
| $\theta_3$ | $U(0, 0.3)$ | Magnitude of seasonality effect |
| $\theta_4$ | $U(0.5, 1)$ | Per person probability of infection scaling factor due to ratio of infectious versus susceptible people in a location |
| $\theta_5$ | $U(0.2, 0.7)$ | Effective infectivity during isolation in household |
| $\theta_6$ | $U(0.1, 0.7)$ | Effective infectivity during isolation in nursing home |
| $\theta_7$ | $U(300, 700)$ | Simulation time (hrs) corresponding to March 27, 2020 |
| $\theta_8$ | $U(0.1, 0.8)$ | Reduction in out of household activities |
| $\theta_9$ | $U(0.01, 0.3)$ | Reduction in transmission due to individual protective behaviors |

Table 3: CityCOVID calibration parameters $\boldsymbol{\theta}$ and priors $\pi(\boldsymbol{\theta})$.

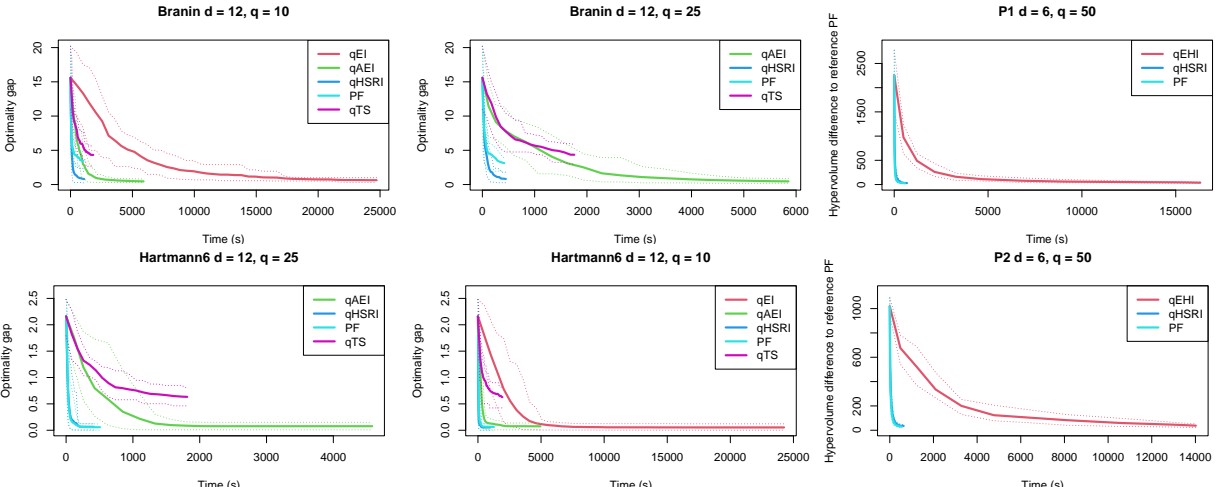

Figure 7: Noiseless results over time. Optimality gap or hypervolume difference over 20 macro-runs is given. Thin dashed lines are 5% and 95% quantiles.

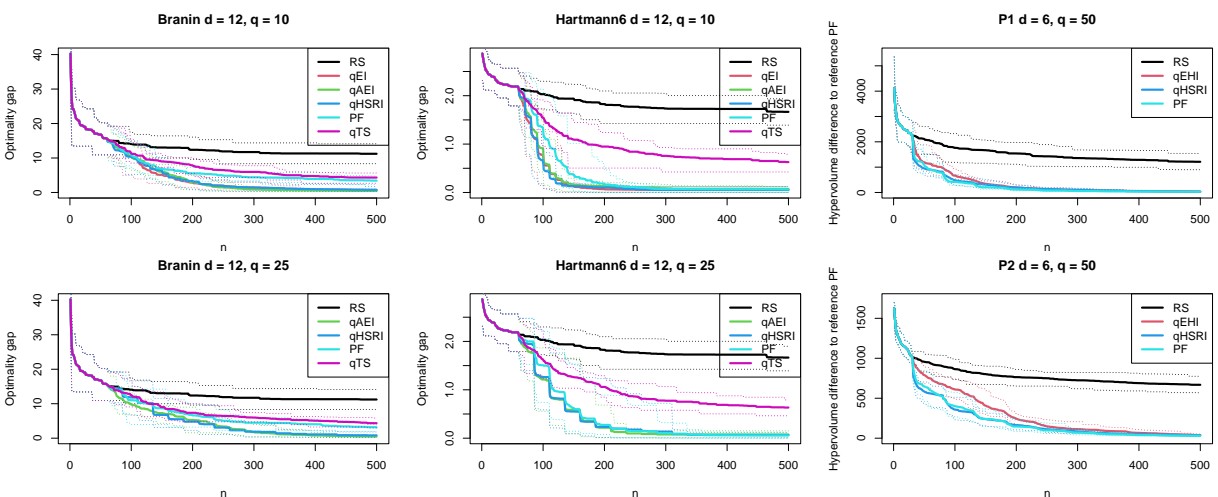

Figure 8: Noiseless results over iterations. Optimality gap or hypervolume difference over 20 macro-runs is given. Thin dashed lines are 5% and 95% quantiles.

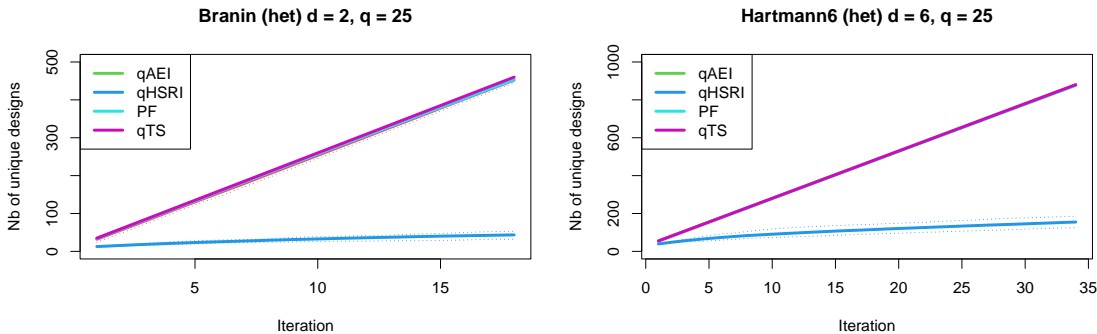

Figure 9: Number of unique designs over iterations for the mono-objective test problems. Thin dashed lines are 5% and 95% quantiles.

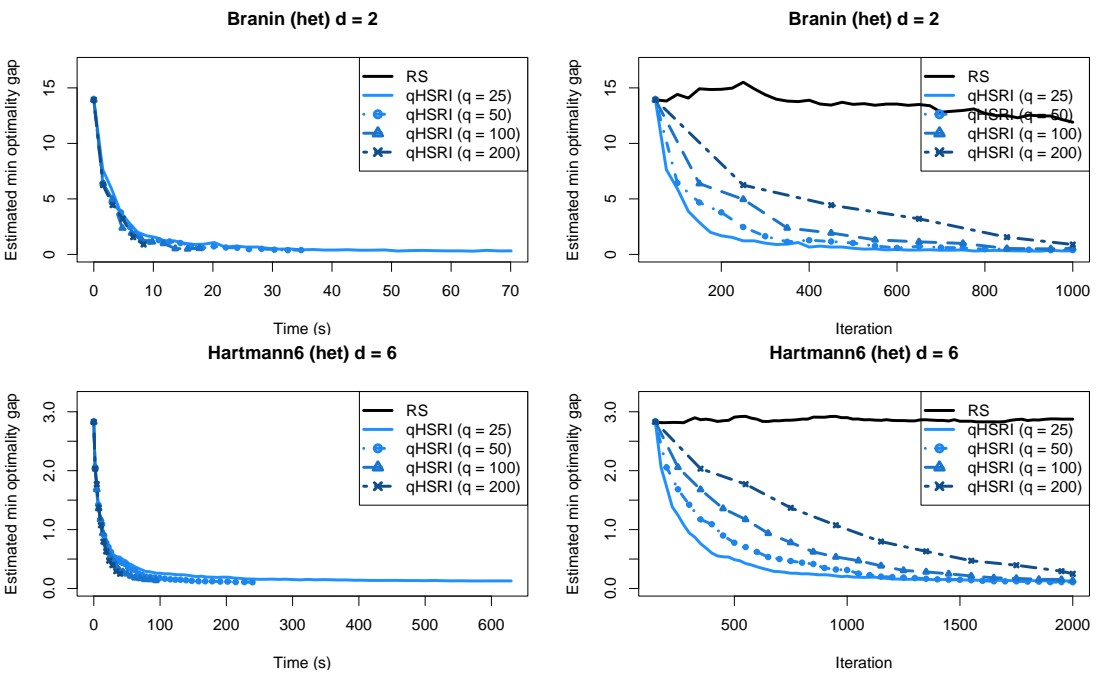

Figure 10: Mono-objective results over iterations and over time. Optimality gap and estimated optimality gap for noisy tests over 60 macro-runs are given.

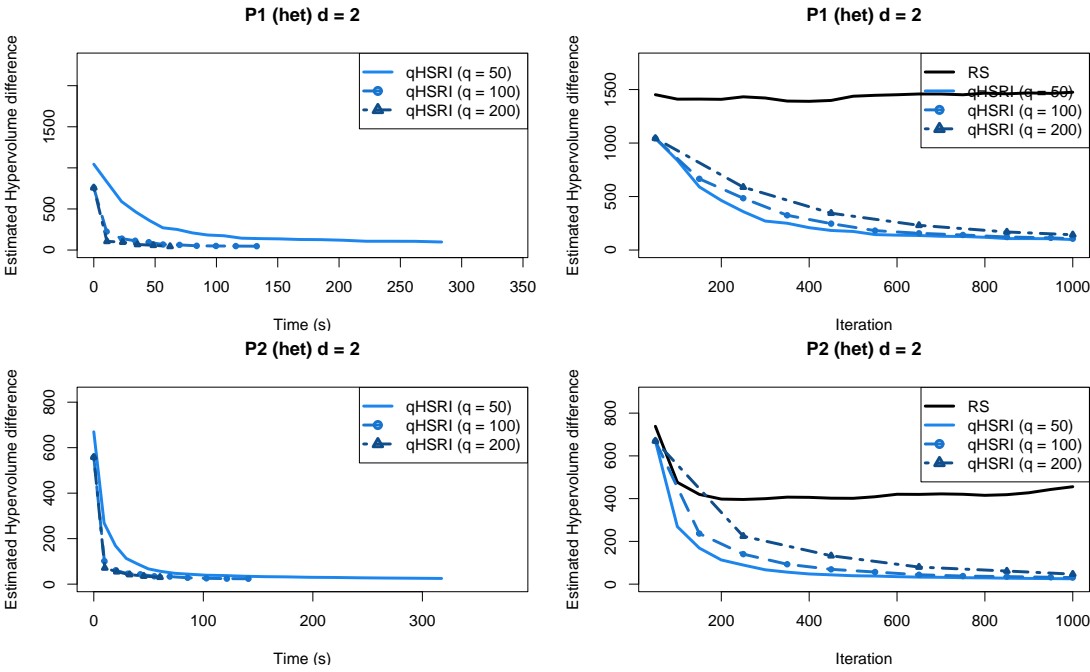

Figure 11: Multi-objective results over time and iterations. Hypervolume difference over a reference Pareto front and its counterpart for the estimated Pareto set for noisy tests over 40 macro-runs are given.

