# OpenReview forum: "A portfolio approach to massively parallel Bayesian optimization"
_TMLR — Rejected by TMLR_

### Review · Reviewer_N7Ge · 2023-02-01

**Summary Of Contributions:**

Bayesian Optimization (BO) is a well-known and sample-efficient technique for optimizing expensive black-box functions. This is important in many applications where no close form of the optimization problem is available, incl. physics and biology but also hyperparameter optimization of ML algorithms. The authors of the paper at hand address the problem of parallel BO, i.e., the efficiency of the optimization process can be improved by querying more than one solution candidate (aka point) at the same time. In particular, they propose BO with a large number of parallel evaluations, e.g., 25, 50 or even 2500 as the authors use in their experiments. Prior parallel BO approaches often do not scale well to this setting because of the complexity of selecting several points. To this end, the authors propose a new variant of hypervolume Sharpe ratio indicator (HSRI) whose overhead scales independently of the number of parallel points to be evaluated. It can further be extended to multi-objective BO. In their experiments, they study the performance of qHSRI on artificial functions and on a real-world benchmark related to COVID.

**Audience:**

Yes

**Claims And Evidence:**

No

**Requested Changes:**

Based on the points raised above, I see the following points as being critical:

* Add a related work section that discusses the differences (pros and cons) of qHSRI and other approaches -- cleaning up the introduction
* Add a better introduction into Subsection 3.3. explaining how HSRI relates to BO
* Add further practical (i.e., non-artificial) benchmarks; e.g., you could use HPO benchmarks from HPOBench

Points to further improve the paper includes:

* Add an empirical comparison against Eriksson et al. 2019
* Explain Algorithm 1 step by step
* Add a scaling study on q (e.g., can be done on one of the artificial functions that allow an arbitrary number of dimensions)


**Strengths And Weaknesses:**

### Strengths

* I agree with the authors regarding the problem setting and that BO is not well equipped for settings with many parallel evaluations. Furthermore, parallelizing many evaluations can be important in some scenarios.
* The scalability regarding overhead of qHSRI is impressive and shows that the authors achieved this goal
* In several experiments, the authors show that qHSRI is very fast and performs reasonably well (see also weaknesses); in particular, it performs regret-wise fairly well but generates much less overhead

### Weaknesses

* First of all, the paper was hard for me to read. In particular, the related work is mixed into the main story all over the place. For example, in the introduction, I wondered for quite some time what the idea and contributions of the paper is.
* Although Section 2 and Subsections 3.1 + 3.2 provide a very nice introduction to the topic, I was lost in Subsection 3.3. It uses a new notation with assets, returns, and so on, and misses to make the connection to the original BO literature. Furthermore, it is not even coherent on its own, e.g., “the outcome is a set of weights” refers to z without being explicit about it.  It took me quite some time to understand this subsection.
* The pseudo-code in Algorithm 1 is not well explained. In fact, the text only says “The pseudo-code of the approach is given in Algorithm 1.”
* I’m missing practical experiments (besides the artificial functions) with some reasonable baselines.
  * Artificial functions such as Branin and Hartmann are known to be a bad proxy for real-world optimization landscapes
  * On the CityCOVID benchmark, only RO and qHSRI are compared with each other
* I missed a comparison against Eriksson et al. 2019 which was mentioned several times and can also be parallelized.
* Why is RO missing in the “Time (s)” plots? It should obviously be the fastest of all methods.
* Although qHSRI is the fastest method, it is not the best wrt optimality gap; in fact, qTS seems to be better and also decently fast.

### Further Comments

* I don’t fully understand the point: “Taking all p standard deviations is possible, but the corresponding objectives are correlated since they increase with the distance to observed design points”. In principle, I agree, but in the assumed setting with heteroscedastic noise, I would argue that one should also assume that the noise level is different for different objectives. Therefore, the std devs would also be different and not perfectly correlated.
* “It could be argued that the qHSRI approach ignores distances in the input space …”. Would it be possible to show or quantify that in some of the experiments?
* I have never read “random search optimization” before. I believe “random search” (w/o optimization) is much more common.
* Please explain why the “Iteration” curves are so bumpy. I would have expected monotonically decreasing curves, similar to the “n” plots
* Why is the heteroscedastic noise in Subsection 4.1. a reasonable choice?
* x-axis label in Figure 5 right is missing

---

> ### Author Response · Authors · 2023-02-23
> **Reply to Reviewer N7Ge**
>
> We are grateful for the helpful comments and suggestions. We reply to these points below.
>
> > First of all, the paper was hard for me to read. In particular, the related work is mixed into the main story all over the place. For example, in the introduction, I wondered for quite some time what the idea and contributions of the paper is.
> > Although Section 2 and Subsections 3.1 + 3.2 provide a very nice introduction to the topic, I was lost in Subsection 3.3. It uses a new notation with assets, returns, and so on, and misses to make the connection to the original BO literature. Furthermore, it is not even coherent on its own, e.g., “the outcome is a set of weights” refers to z without being explicit about it. It took me quite some time to understand this subsection.
>
> We have carefully taken into account these comments. See our detailed reply in the general response.
>
> > The pseudo-code in Algorithm 1 is not well explained. In fact, the text only says “The pseudo-code of the approach is given in Algorithm 1.”
>
> We have added the following paragraph to detail Algorithm 1.
>
> ``The pseudo-code of the approach is given in Algorithm 2.
> The first step is to identify $s$ designs on the mean vs.\ standard deviation Pareto set. In the deterministic case $s$ must be at least equal to $q$, while with noise and replication it can be lower. Population based evolutionary algorithms can be used here, with a sufficiently large population. Once these $s$ candidates are identified, dominated points can be filtered out as HSRI only selects non-dominated solutions. Points with low probability of improvement (or with low probability of being non-dominated) can be removed as well. In the noisy case, the variance reduction serves as an additional objective for the computation of HSRI. It is integrated afterwards as it is not directly related to the identification of the exploration-exploitation tradeoff surface. Computing qHSRI then involves computing $\mathbf{r}$ and $\mathbf{Q}$ before solving the corresponding QP problem (1). Designs from $\mathbf{X}_s$ are selected based on $\mathbf{z}^*$: either by taking the $q$ designs having the largest weights $z_i$, or computing $\gamma$ to obtain an allocation, which can include replicates."
>
> > Artificial functions such as Branin and Hartmann are known to be a bad proxy for real-world optimization landscapes
>
>  Branin and Hartman remain standard benchmarks for BO. As for the CityCOVID benchmark, no other alternative is reasonable due to the value of $q$.
>
> > Why is RO missing in the “Time (s)” plots? It should obviously be the fastest of all methods.
>
> On the synthetic functions the evaluation time is negligible for RO. The corresponding (bad) performance is visible on the figures per iteration that are next to them. On the CityCOVID benchmark, evaluation times do not vary much hence a plot with respect to time would look similar.
>
> > Although qHSRI is the fastest method, it is not the best wrt optimality gap; in fact, qTS seems to be better and also decently fast.
>
> Indeed, qHSRI is not always the best method iteration-wise, for the moderate values of $q$ where comparison is possible. But here our primary goal is time to solution. Also qTS only shows a good performance for Hartmann6 (we added a comment in the text), it performs bad in the deterministic setup and is not suited for multi-objective optimization.
>
> > I don’t fully understand the point: “Taking all $p$ standard deviations is possible, but the corresponding objectives are correlated since they increase with the distance to observed design points”. In principle, I agree, but in the assumed setting with heteroscedastic noise, I would argue that one should also assume that the noise level is different for different objectives. Therefore, the std devs would also be different and not perfectly correlated.
>
> We now detail more this point:
>
> ``Even with different hyperparameters, preliminary tests showed that the averaged predictive standard deviation is sufficient compared to the additional difficulty of handling more objectives.''
>
> Concerning heteroscedastic noise, this aspect is taken into account by the additional variance reduction objective. We have also added:
>
> ``In the noisy case, the variance reduction serves as an additional objective for the computation of HSRI. It is integrated afterwards as it is not directly related to the identification of the exploration-exploitation tradeoff surface.''
>
> > “It could be argued that the qHSRI approach ignores distances in the input space …”. Would it be possible to show or quantify that in some of the experiments?
>
> As we now describe in the text, this is related to the shape and extent of the Pareto set corresponding to the exploration-exploitation tradeoff. For instance, it can be disconnected or not depending on the iterations and test problem, making it hard to quantify.

---

> > ### Author Response · Authors · 2023-02-23
> > **Reply to Reviewer N7Ge (continued)**
> >
> > > I have never read “random search optimization” before. I believe “random search” (w/o optimization) is much more common.
> >
> > We have changed random search optimization to random search accordingly.
> >
> > > Please explain why the “Iteration” curves are so bumpy. I would have expected monotonically decreasing curves, similar to the “n” plots
> >
> > The “n” plots can be misleading when there is noise, they do not take into account the model estimation about the minimum at each stage, which can be obscured by noise. We added:
> >
> > ``the optimality gap only assesses whether a design has been sampled close to an optimal one, not if it has been correctly predicted.''
> >
> > > Why is the heteroscedastic noise in Subsection 4.1. a reasonable choice?
> >
> > This choice was made to use standard functions for BO, with a large and varying noise variance while allowing to get results for the various baselines in reasonable times and number of iterations.
> >
> > > Add an empirical comparison against Eriksson et al. 2019
> >
> > TurBO indeed shows good results for moderate batch sizes in high-dimension, but it does not handle replication that is beneficial for large signal to noise ratio. Further work would be needed to adapt it to this context.
> >
> > > Add a scaling study on $q$ (e.g., can be done on one of the artificial functions that allow an arbitrary number of dimensions)
> >
> > The results of this study have been added in Appendix.

---

> > > ### Comment · Reviewer_N7Ge · 2023-02-25
> > > **Thanks for the Revision**
> > >
> > > Thank you for the revision and all the replies. The paper has substantially improved. Nevertheless, I'm still not entirely convinced by the experiments -- they are better explained, but the baselines are still weak and the limited number of benchmarks does not provide sufficient evidence.

---

### Review · Reviewer_dXo3 · 2023-02-12

**Summary Of Contributions:**

This paper tackles large batch (or parallel) multi-objective Bayesian optimization (BO), a problem that is known. The authors propose an approach based on solving a "portfolio optimization" problem once a set of points on the exploration-exploitation Pareto front (where the objectives are posterior means and posterior variance) are identified. Using this method, the authors show large (order of magnitude) improvements over existing methods.

**Audience:**

Yes

**Claims And Evidence:**

No

**Requested Changes:**

I think all of the clarity related issues above are necessary for publication of this paper.

**Strengths And Weaknesses:**

Strengths:
- This is an important problem and currently there is, to my knowledge, not a satisfactory method for performing extremely large batch BO, besides optimizing sequentially.

Weaknesses:
- The major weakness, in my view, is that the paper needs improved clarity and a more precise description of the method. There are quite a few points that were unclear to me, listed in the next few bullet points.
- In Algorithm 1, what is `s` in X_s? It seems like this is the number of points to use to approximate the exploration-exploitation Pareto frontier. But should `s` be smaller or larger than `q`? If `q` is really large, and we want to find unique points, then `s` would also have to be really large. In that case, the first line of Algorithm 1 becomes a difficult MO problem. I would be interested in reading some discussion on this point.
- I did not fully comprehend why we should not use all of the posterior standard deviations, especially for the MO case (the text in the paper mentions that there is correlation). Could this be explained more clearly?
- The paper describes a two-step approach of first finding a Pareto frontier and then using a Sharpe ratio optimization problem. Is there a way to write this as a single joint optimization problem that sheds more light on exactly what the end goal is? In other words, it would be nice to be able to write down the acquisition function using some simple equations that illustrate the main idea.
- The novelty of this paper beyond existing work (e.g., Guerreiro and Fonseca (2016), Gupta et al. (2018), Yevseyeva et al. (2014)) should be more clearly discussed. My sense is that the current paper adapts these previous methods to the batch BO setting.

---

### Review · Reviewer_uaxT · 2023-02-14

**Summary Of Contributions:**

In each iteration, Bayesian optimization samples new candidate via an acquisition function, which is based on a probabilistic model of the objective function.
The acquisition function quantifies the trade-off between exploration in the search spaces where the model is uncertain and exploitation where the predictive mean of the model is low.
This trade-off can be viewed as multi-objective problem, and popular acquisition functions, such as Expected improvement or UCB as scalarised formulation of this problem.

The main contribution of the paper is a new batch selection method for synchronous parallel Bayesian optimization, where, in each iteration, candidates are samples in batches rather than sequentially.
The computational cost of selecting a batch of candidates for existing methods increases with the batch size, which can become the computational bottleneck if the objective function is not too expensive to evaluate.
The paper proposes to use the hypervolume Sharpe ratio indicator (HSRI) to select a batch where the computational costs increases linearly with the number of objectives instead.

**Audience:**

Yes

**Broader Impact Concerns:**

I do not see any broader ethical concerns of this work.

**Claims And Evidence:**

No

**Requested Changes:**

- Extend Section 2 to introduce Bayesian optimization and give a high level overview of the basic algorithm

- I'd recommend to evaluate the proposed method on surrogate benchmarks from the literature which require the same computation cost as synthetic benchmarks but resemble practical use-cases. See for example:

HPOBench: A Collection of Reproducible Multi-Fidelity Benchmark Problems for HPO
 Katharina Eggensperger, Philipp Müller, Neeratyoy Mallik, Matthias Feurer, René Sass, Aaron Klein, Noor Awad, Marius Lindauer, Frank Hutter

- Consider asynchronous Bayesian optimization methods as baselines


- Further clarifications:

  -   What is the difference between estimated HV difference and HV difference?

   - Revise Section 3.3: it's hard to understand the mapping of different components of HSRI to batch Bayesian optimization.



**Strengths And Weaknesses:**

## Strengths


It seems quite sensible to explicitly model the exploration / exploitation trade-off via the predictive mean and standard deviation of the model and to select the next batch by sampling Pareto optimal points.


## Weaknesses


### Motivation of the paper

I am a bit sceptical about the motivation of the paper.  Bayesian optimization seems to be particularly useful when the objective function is expensive-to-evaluation, since it is often more sample efficient than other approaches, such as random-search based methods or evolutionary algorithms. With this in mind, I am wondering how relevant is the scenario of such large batch setting given that it only requires a few seconds per evaluation with q=100?
Also, it seems that the large body of work on asynchronous Bayesian optimization does not suffer from this problem and might be more suitable for this setting.



### Writing

Overall, I found the paper rather confusingly written and I do not think that it is very accessible for readers not deeply familiar with the subject. For example, while Gaussian processes are described in Section 2, Bayesian optimization is not formally introduced. For example, the acquisition function is not properly defined; the Bayesian optimization loop is not described. Instead, the paper immediately jumps into Batch Bayesian optimization.
Furthermore, the paper contains a lot of jargon (Hypervolume) and many terms are not mathematically well defined (Probability of improvement, Pareto set, etc).



### Weak empirical evaluation

I have several concerns about the empirical evaluation of the paper:

 - The choice to use the same hyperparameters for the GP for each objective seem questionable. I expect especially the length scale to change with each objective function.

 - The paper presents only single run on lunar landar which seems to exhibit substantially amount of noise. This seems in insufficient to draw any reliable conclusions.

 - Except the CityCovid datasets, the paper only considers synthetic problem, which makes it hard to judge how relevant the proposed gains are in practice.

---

### Decision · Action_Editors · 2023-03-13

**Recommendation:** Reject

**Comment:**



Bayesian optimisation has to deal with the exploration-exploitation trade-off---do we exploit the current knowledge of the unknown objective function to find the optimum or do we spend more resources on learning about the unknown objective function? Existing acquisition functions commonly choose a-priori a trade-off between exploration and exploitation. In this paper, like Gupta et al 2018, the authors first determine the Pareto front representing the trade-off and propose to then select points from
it a-posteriori in a batch setting using a portfolio selection criterion. This is in contrast to Gupta et al 2018 who use random (sub)-sampling.

There is consensus among the reviewers that the paper is not yet ready for publication. The main issues to improve on are:

(1) The empirical performance and evaluation.

(2) The clarity of the writing.

For both points, the reviewers made a number of suggestions (e.g. benchmarks from [1-3] for point (1)) that were not sufficiently incorporated in the revision. I recommend to take them at heart to improve future versions of the paper.

Moreover, based on the write-up, it seems to me that the paper does not present results for Gupta et 2018 (denoted as PF in the paper) in case of the Lunar and CityCovid problem, or discuss why it's not feasible. This is rather surprising because the presented work immediately builds on theirs and we needed to see clear evidence for the relative added-value. Relatedly, the paper needed to more prominently give credit and discuss differences to their predecessors (Guerreiro and Fonseca (2016), Gupta et al.
(2018), Yevseyeva et al. (2014)).

[1] HPOBench: A Collection of Reproducible Multi-Fidelity Benchmark Problems for
HPO Katharina Eggensperger, Philipp Müller, Neeratyoy Mallik, Matthias Feurer,
René Sass, Aaron Klein, Noor Awad, Marius Lindauer, Frank Hutter

[2] YAHPO Gym - An Efficient Multi-Objective Multi-Fidelity Benchmark for Hyperparameter Optimization Florian Pfisterer, Lennart Schneider, Julia Moosbauer, Martin Binder, Bernd Bischl

[3] Scalable Global Optimization via Local Bayesian Optimization D Eriksson, M Pearce, J Gardner, RD Turner, M Poloczek

**Audience:**

The topic of the paper is of interest to some of TMLR's audience. However, the clarity of the writing remains insufficient and needed to be improved to make this work more accessible to the reader.


**Claims And Evidence:**

While the reviewers appreciate the effort that went into improving the paper, there is consensus that the paper does not provide sufficient empirical evidence for the claimed benefits of the proposed method. In particular, the baselines are too weak and the chosen benchmarks do not provide sufficient evidence.